# Printed Eddy Current Testing Sensors: Toward Structural Health Monitoring Applications

**DOI:** 10.3390/s23198345

**Published:** 2023-10-09

**Authors:** Eliott Brun, Pierre-Jean Cottinet, Arnaud Pelletier, Benjamin Ducharne

**Affiliations:** 1LGEF, INSA-Lyon, EA682, University Lyon, 69621 Villeurbanne, France; eliott.brun@insa-lyon.fr (E.B.); pierre-jean.cottinet@insa-lyon.fr (P.-J.C.); 2CMPhy, 26 Rue Paul Sabatier, 71530 Crissey, France; arnaud.pelletier@cmphy.fr; 3ELyTMaX UMI 3757, CNRS, Univ. Lyon, INSA Lyon, Centrale Lyon, Université Claude Bernard Lyon 1, Tohoku University, Sendai 980-8577, Japan

**Keywords:** non-destructive testing, dispenser-printed sensor, screen-printed sensor, embedded sensor, noninvasive sensor

## Abstract

Reliable measurements in structural health monitoring mean for the instrumentation to be set in perfect reproducible conditions. The solution described in this study consists of printing the sensors directly on the parts to be controlled. This method solves the reproducibility issue, limits human error, and can be used in confined or hazardous environments. This work was limited to eddy current testing, but the settings and conclusions are transposable to any non-destructive testing methods (ultrasounds, etc.). The first salve of tests was run to establish the best dielectric and conductive ink combination. The Dupont ink combination gave the best performances. Then, the dispenser- and the screen-printing methods were carried out to print flat spiral coils on flexible substrates. The resulting sensors were compared to flex-printed circuit boards (PCB-flex) using copper for the electrical circuit. The conductive ink methods were revealed to be just as efficient. The last stage of this work consisted of printing sensors on solid parts. For this, 20-turn spiral coils were printed on 3 mm thick stainless-steel plates. The permanent sensors showed good sensibility in the same range as the portative ones, demonstrating the method’s feasibility.

## 1. Introduction

Most existing Non-destructive Testing (NDT) methods and equipment generally come in the form of measurement sensors intended to be used manually by on-site operators. They are mainly designed for end-of-production line controls and to detect macroscopic defects such as cracks, porosities, inclusions, and lack of materials. They are also used in maintenance services to control the creation of defects and their incidence after a certain number of operating cycles [1,2,3].

Even if they were not designed for this, these control methods could also provide information regarding the health of the material and its evolution over time. However, the cost and size of these measurement methods and the need for an operator to carry out these controls can sometimes hinder their practical use and overall development in the framework of material and structural health monitoring [4,5,6].

In addition, some structures requiring periodic checks are complicated to control due to their environment. The leading causes of these difficulties can be:–A confined environment where operators can hardly reach the parts that need to be controlled, resulting in shutdowns for maintenance that are potentially lengthy and expensive.–To work at a height like civil engineering structures, where access is complex and requires specialized operators and equipment.–Dangerous environments characterized by high temperatures, pressures, or radiation.

Finally, ensuring reliable measurements in structural health monitoring requires the instrumentation to be placed under consistently reproducible conditions.

Eddy Current Testing (ECT) is an NDT method developed for inspecting electrically conductive materials such as metals or carbon composites [7,8,9]. In its most basic form, the single-element ECT probe (a coil of conductive wire) is excited with an alternating electrical current. The electrical current produces an alternating magnetic field in the vicinity of the ECT probe, oscillating at the same frequency. When the coil approaches a part to be controlled, currents are induced in the material under the action of this magnetic field: the eddy currents [10].

Variations in the electrical conductivity and magnetic permeability of the test object or the presence of defects cause a change in the eddy current distribution. This variation results in a change in phase and amplitude that can be detected by monitoring the coil impedance along with the experimental process and giving a telltale sign of the presence of defects.

ECT architecture is simple enough to be considered for miniaturization and development in the frame of a Structural Health Monitoring (SHM) design [11,12,13]. More specifically for the sensor coil, this miniaturization is possible using planar coils made on flex Printed Circuit Boards (PCB) or even printed, through conductive inks, directly on the parts to be controlled. Printing methods can be used to obtain noninvasive sensors of reduced thickness and minimal influence on the tested components [14,15,16,17]. They can be implanted permanently on tested parts (Figure 1) and used to conduct testing campaigns in confined or dangerous environments. Eventually, printed sensors can also be exploited for inspecting non-flat surfaces (welds [18], ball joints, etc.), which are impossible to control with conventional methods or where bonding a sensor would be unrealistic. 

Miniaturized and printed ECT coils can be built apart and used as a conventional ECT tool (manually by an operator). They can also be printed directly on the component to be controlled (Figure 1) during manufacturing and serve as tools for SHM and analysis. This perspective leads to “intelligent” structures and products capable of self-diagnosing and communicating with outside maintenance services [19].

The method described in this paper involves directly printing sensors on the component surface slated for monitoring. This innovative technique effectively resolves the issue of reproducibility, reduces the potential for human error, and is applicable even in confined, hazardous environments. While this study specifically focused on eddy current testing, the configurations and findings can be extrapolated to any NDT methods (ultrasounds, etc.). Printed technology has recently been used in the NDT framework (in [20], an ink-jet printer is used to print eddy current sensors, and in [21], flexible printed coil arrays are deployed to test an NDT wireless technology) but never to print sensors directly on the part to be controlled.

This study aims to determine the printed sensor coils’ “ideal” design and integration. We will answer questions associated with the capability of this new generation of sensors in an NDT and SHM context. We want to determine the key parameters leading to the best performances and the interactions between these parameters. The manuscript is organized as follows: Section 2 gives essential design rules. Section 3 explains the printed sensors’ design and construction. Two printing techniques are described and tested on flexible substrates: dispenser and screen printing. The flex Printed Circuit Board (PCB) using copper is also described as it will be used to compare and validate the conductive ink-printed sensors. Section 4 is dedicated to the characterization of the developed sensors. Then, the best configuration (inks and printing technique) is implemented on a mechanical part and tested in NDT conditions. Conclusions and discussions follow in Section 5.

## 2. Design Rules

The leading objective of this study is the development of printed sensors and their use for SHM and defect detection. To reach our goal, we need to design sensors with specific geometry. In the particular case of ECT, flat spiral coils are recommended. In a classical NDT process, the operator will perform manual scans along the part to be tested to locate potential defects precisely. The same result can only be reached with permanently printed sensors by fully paving the part’s surface to be controlled (Figure 1). The pavement resolution must be as fine as possible to maximize the detection efficiency. It is inconceivable to use techniques picked up in the framework of microelectronics to pave large industrial parts. Therefore, the maximum resolution becomes inherent to the manufacturing technologies selected for the printing process. We opted for a sensor surface of about 20 × 20 mm^2^, which seemed an acceptable resolution for most mechanical parts targeted as potential candidates for periodic SHM controls. Such a surface allows 10- to 20-turn coils to be easily printed. The PCB technology has a better resolution, and a 50-turn coil can be obtained for the same surface. For ECT sensors of improved performances, the coil quality factor Q (Equation (1)) has to be maximized [22,23]:(1)$$Q=\frac{Z^{\prime \prime }}{Z^{\prime }}$$

Here, Z = Z′ + jZ″ is the coil impedance, Z′ = R, and Z″ = Lω are the impedance’s real and imaginary parts. R is the coil resistance, L is the coil inductance, and ω = 2π*f* the angular frequency. A good Q implies maximizing L vs. R. The easiest way to increase L is to increase the number of turns. However, such a change means geometrical variations which are deleterious to other characteristics. Different options can be listed to increase the number of turns (see Figure 2a for illustration):–Increase the sensor’s surface, but this solution will decrease the paving resolution (Figure 2(a-1)).–Keep with the same surface but decrease the size of the conductive tracks and add new turns to the original design. Still, this solution will inherently increase the resistance (Figure 2(a-2)).–Decrease the distance between conductive tracks, but the risk is to add parasitic capacities to the original design [24,25] (Figure 2(a-3)).–Opt for a multilayer coil [26], but the printing difficulty increases exponentially due to the superimposition of conductive and dielectric layers (Figure 2(a-4)).

**Figure 2 sensors-23-08345-f002:**
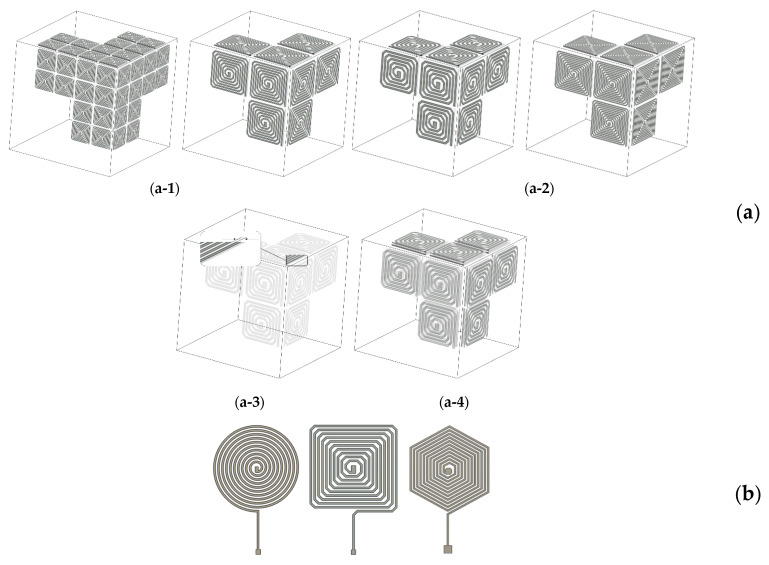
(**a**) Illustration of the different options to increase a planar coil’s number of turns. (**a-1**) Larger sensor’s surface → lower paving resolution. (**a-2**) Same surface, larger turns number → higher in series resistance. (**a-3**) Lower conductive track distance. (**a-4**) Multilayer coil resistance. (**b**) Circular, square, and hexagonal 10-turn planar coil.

The value of R constitutes another degree of freedom, but its reduction implies geometrical changes affecting the characteristics of the printed sensor. Different options can be listed as well for a reduction of R:–Increase the conductive track width but potentially be faced with parasitic capacities [27].–Decrease the total length of the conductive tracks, which means fewer turns and smaller inductance.

Eventually, a compromise has to be made to reach the best performance. A wide range of shapes can be envisaged for the planar spiral coil (Figure 2b). The most common are circular, square, or rectangular. Each of these shapes has pros and cons. The targeted defect’s nature and size help select the best one. The circular shape is, for instance, recommended for detecting micro-cracks [28]. The issue with sensors made of straight lines (square, rectangular, etc.) comes from cracks growing parallel to those lines and inducing limited impedance variations. In the SHM context, where whole-surface paving is required, circular sensors produce “blind” areas where nothing can be detected. Hexagonal coils constitute a possible compromise (Figure 2b). Even if straight lines are still needed, they are of limited length. Full-surface paving is possible, and the level of crack detection is significantly improved.

## 3. Printed Sensors Development

Three manufacturing techniques were tested for the development of our printed coils: dispenser printing, screen printing, and PCB-Flex. The dispenser- and screen-printing techniques involve dielectric and conductive inks and allow the manufacture of the sensor directly on the part to be tested. Since ECT requires the materials to be controlled to be conductive, it is impossible to print right on a metallic part, and a layer of dielectric insulation has to be implemented in the first stage. Then comes the conductive ink flat spiral coil layer, followed by another dielectric layer to protect the sensor from the external environment. Finally, a last conductive ink layer is required to connect the center of the spiral coil and enable electrical measurements. Figure 3 illustrates the printing process and the different layers. The bottom right sketch gives an overview of the sensor in its final stage. The dimension details are given in Figure 4.

PCB-Flex sensors are less versatile. They take the form of polyimide film that can be bonded on the parts to be controlled. This solution is attractive as sensors can be printed in large series independently of the mechanical part to be controlled. Still, bonding is not ideal, as universal adhesives contain chemicals that may suffer hydrolytic degradation due to external conditions and long storage times. Adhesive bonds exposed to elevated temperatures well within their operating condition often fail if they are cooled too quickly at the low end of their thermal ranges [29].

### 3.1. Dispenser Printing

A dispenser-printing system consists of an ink syringe that deposits ink over a substrate. Inks with a wide range of viscosities can be used. The ink can be printed continuously, leading to filaments or through successive drops (Figure 5). Precise pressure control in the ink barrel allows switching from the continuous to the discontinuous printing modes [30,31,32,33].

The needle diameter can range from 0.10 to 400 mm. It is made of pulled glass capillaries for small diameters and stainless steel for others. The pressure depends on the ink’s viscoelastic and the needle’s diameter.

In this study, the dispensing printing system used was a 3D printer from Hyrel 3D (Atlanta, GA, USA), referred to as Hyrel System 30M (shown in Figure 6). The 3D printer is associated with a print head: VOL-25. A dedicated software named Repetrel controls the whole system in a dispensing printing mode. This printing head comprises an anodized aluminum syringe whose piston displacement is controlled by an endless screw. The needle is assembled on specific support, including a stepper motor for precisely setting the couple screw/piston motions. The width of the deposited tracks depends on the needle diameter. This study obtained the best results (10-turn spiral coil in a 150 mm^2^ area) with a 0.33 mm diameter needle.

Repetrel gives access to multiple parameters, including the printer head displacement speed and the step motor rotational speed associated with the ink flow. The setting of these parameters constitutes the principal difficulty in obtaining quality printing.

In this work, the sensors were designed using the Computer-Aided Design (CAD) software Fusion 360 (Autodesk, Mill Valley, CA, USA) and saved in an .stl file before transfer to the Hyrel 3D printer.

### 3.2. Inks Selection

Preparing inks for dispenser printing appears more complex than other printing methods. Inks combine conductive particles and suitable solvents. Due to the difference in the densities and the large particle size distribution, dispenser-printing inks can easily clog the needle during the printing session. In this study, we have searched for conductive and dielectric inks that can withstand the most extreme working conditions (according to our industrial partners): up to a temperature of 200 °C. A vast range of technical data were compared, and combinations of inks were selected:–An epoxy resin-based ink for the dielectric layer and an epoxy resin/carbon particles ink for the conductive one, both from Creative Material (Albany, NJ, USA).–A polyimide-based ink for the dielectric layer and a polyimide/silver particles ink for the conductive one, both from Dupont (Wilmington, NC, USA).–A low-cost alternative: a silver lacquer (RS Pro conductive paint, London, UK) for the conductive layer and a high-temperature silicon-based paint for the dielectric one (MOTIP heat-resistant paint, Valkenburg, The Netherlands).

Multiple printability tests and temperature resistance revealed the combination from Dupont as the most suitable one. The carbon/epoxy combination was discarded for its limited conductivity and temperature dependency, unsuitable to our needs. The low-cost solution was rapidly discarded as the silver lacquer clogged the printing needle. Figure 7b shows 10- and 20-turn spiral coil sensors obtained with the dispenser-printing technique and the Dupont ink combinations. Endurance tests on the evolution in time of the selected combination of inks are still ongoing. The results of these tests will be published later in another article. Even if the final objective of this study is to print permanent sensors, we decided to print our first batch of sensors on a 50 μm Kapton sheet. The features of the printed techniques can be evaluated by comparing the sensors in the same experimental conditions (same substrate). The printed sensors can be moved easily to test different natures of material. Kapton is “cheap” and characterized by limited surface roughness, easing the printing process. Kapton is also highly flexible, and the resulting sensors can be tested on a vast range of geometry. Finally, Kapton has excellent heat resistance, providing us with an additional degree of freedom for the test conditions.

### 3.3. Screen Printing

Screen printing is a technique where a mesh transfers ink onto a substrate, except in areas made impermeable by a blocking stencil. A squeegee is moved across the screen to fill the open-mesh apertures with ink. A reverse stroke causes the screen to touch the substrate momentarily along a contact line. This causes the ink to wet the substrate and be pulled out of the mesh apertures as the screen springs back after the blade has passed [34].

All ink combinations described above can be used successfully for screen printing. Still, we limited the test to Dupont’s combination, which gave the best thermal resistance results. Screen printing offers many advantages, including:–a good reproducibility;–a higher resolution that leads to more turns within a specific area.

Difficulties can also be associated with screen printing, including the high wettability of the conductive ink on specific substrates (Kapton polyimide film is one of them). The conductive ink can potentially be flowing and locally modify the sensor geometry. Figure 7a shows a 20-turn spiral coil sensor obtained using the screen-printing technique and the combination of Dupont inks. For comparison purposes, the substrate is again a 50 μm Kapton sheet.

### 3.4. Printed Circuit Boards-Flex: PCB-Flex

The PCB flex technology involves engraving a copper conductive layer on a flexible substrate. While standard PCBs use a fiberglass or metal base, PCB flex substrates are made of a flexible polymer, resulting in a flexible circuit board that keeps its functionality even in a folded configuration.

Most of the time, polyimide films are used. Polyimide exhibits ideal thermo-mechanical properties, including remaining flexible even after thermosetting.

In the standard process, once the engraving stage is done, a second polyimide film is deposited to encapsulate and isolate the circuit from the external environment. In this work, all PCB flex sensors were designed using the CAD Altium (San Diego, CA, USA) Designer software (3.5.0.17). Then, the manufacturing process was assigned to PCB Way, Shenzhen JDB Technology (Shenzhen, China). Figure 7c shows 10- and 20-turn spiral coil PCB flex sensors.

## 4. Printed Sensors Characterization

### 4.1. Detection Capability of the Sensors Printed on a Kapton Substrate

A precise characterization of the printed sensors is mandatory to confirm their viability and define their detection capability. This experimental characterization comprises two stages:

The first stage involves placing the tested sensors far from conductive or magnetic parts to avoid eddy current generation and/or undesired interaction. In our case, the evolution of the electrical impedance was measured with an Agilent (Santa Clara, CA, USA) E5061B network LCR analyzer. The tested frequency *f* was swept up from 10 kHz to 5 MHz. Then, for each frequency, the coil resistance, coil inductance, and quality factor were returned and used to compare the sensors. All tested sensors had the same geometry: 10-turn spiral coils, 20 × 20 mm^2^ surface area. Table 1 gives the resulting data for the three types of sensors.

The following conclusions can be drawn from the Table 1 results:–The inductance variations are limited from one sensor to the other, confirming the coil geometry and the number of turns as the only parameters influencing the inductance value.–The resistances of the dispenser- and screen-printing techniques are close, around ten times higher than the PCB ones. Such a difference was forecasted and is mainly due to the relatively limited conductivity of the Dupont ink (A post-processing analysis revealed σ = 2 × 10^6^ S·m^−1^ for the Dupont ink and around σ = 5 × 10^7^ S·m^−1^ for the flex PCB’s copper). The difference between the coil resistances of the dispenser- and the screen-printed sensors is mainly due to some geometrical imperfections of the dispensed one.–The dispenser- and the screen-printed sensors exhibit very close features. Both methods can be equally used for the permanent sensor.

Other geometries have been tested, including sensors with more turns (Table 2). Applying more turns means more inductance. Still, the quality factor remains relatively unchanged as the inductance variation is compensated by the resistance one.

A change in the number of turns modifies the coil resonant frequency (from 18.8 MHz for the 10 turns to 8.2 MHz for the 20 turns), which sets the range of use. Finally, comparative tests between the different shapes of coils were conducted. They did not reveal any significant difference. The similarity of the intrinsic performances was concluded.

The second stage involves replicating the same tests but placing the sensors near a conductive and possibly magnetic specimen. Eddy currents are supposed to be generated in the conductive parts and modify the sensor coil impedances [7,35,36,37,38]. Different natures of specimens were tested, and conclusions were drawn regarding the capability of the printed sensors to discriminate one type of material from the other. All tested specimens were explicitly selected as model materials classically tested in the industry. Those include carbon composites, stainless-steel parts, and ferritic-steel parts. Table 3 depicts the precise list of all tested specimens:

Figure 8 shows all tested specimens’ measurements plotted on the same impedance plane and for different levels of frequencies (*f* Є [1 × 10^5^–5 × 10^6^] Hz). The sensor was a 20-turn spiral coil. The left-hand side of Figure 8 was measured with a dispenser-printed sensor and the Dupont ink combination, and the right-hand side with the PCB. The screen-printed sensor measurements are not displayed here for space limitation but are similar to the dispenser-printed ones.

The Figure 9 histograms provide interesting comparisons and insights into the interactions between the sensors and the electrical conducting materials. As predicted, larger real impedance parts are obtained for ferritic steels characterized by higher magnetic permeabilities (stainless and ferritic steels exhibit the same range of electrical conductivities). About the imaginary parts: slight variation is observed for the diamagnetic carbon specimens. For an accurate comparison of the ferritic and stainless steels, Kirchhoff’s laws can be applied to calculate the effective impedance during the ECT test [39,40]:(2)$$R=R_{c}+\frac{\omega^{2}M2}{R_{S}^{2}+(\omega L_{S})2}R_{S}$$
(3)$$L=L_{c}-\frac{\omega^{2}M2}{R_{S}^{2}+(\omega L_{S})2}L_{S}$$

R_c_ and L_c_ are the coil series resistance and inductance in a vacuum environment, respectively. R_s_ and L_s_ are those of the substrate, and M is the mutual inductance between the sensor and the substrate. In the high-frequency range, R_s_ is weak vs. ωL_s_, and the following approximation can be made:(4)$$L\propto L_{c}-\frac{M2}{L_{s}}$$

In such conditions, the inductance value gives a direct image of the substrate permeability.

### 4.2. Sensors Implementation on Mechanical Parts

The last stage of this work consists in verifying the method’s viability by printing the sensors on mechanical parts. For this, 20-turn spiral coils were dispenser-printed on 3 mm thick, stainless steel blocks. The Dupont inks combination was used. One of these resulting sensors is depicted below.

Six stainless steel blocks were instrumented to test the reproducibility of the measurements, and two blocks were left untouched for the same reasons. The reproducibility of the instrumented blocks was confirmed. Then, we compared the permanent sensor’s electromagnetic responses to those of the portable ones depicted in Figure 7 and of similar features (same geometry, same ink combination). Two mobile sensors were positioned on the virgin blocks’ top surface and monitored their electromagnetic responses. Again, they gave very similar answers. Figure 10 shows the comparison between the portable and the permanent sensors answers. The slight difference in the amplitude of IZI can be attributed to the variability in the printing process, resulting in minor variations of the static resistance (259 Ω for the permanent sensors and only 242 Ω for the portable ones). At 1 MHz, both coil inductances remain in the same range (≈3.5 × 10^−6^ H), confirming the permanent printing solution’s feasibility. Another validation stage was performed by drilling a small-sized hole (Ø 0.35 mm, length 2 mm) in the bottom face of an instrumented part and acting as a potential defect, as illustrated in Figure 11.

A new experimental campaign was run in Figure 9 experimental conditions, and the damaged part’s resulting impedance variations were compared to those of valid ones. Variations were observed in the complex plan diagrams depicted in Figure 12. To be more accurate, we observed a shift in the real part of the coil’s impedance of about 2 Ω for the low frequencies and up to 2.5 Ω for high frequencies. This shift in the sensor response clearly shows that the drilled hole is detected, confirming the reliability of the printed sensors in experimental conditions close to those of an industrial environment.

As a final test, the mobile sensors were tested with different conditioning electronics, including commercialized ones [41,42]. These tests were carried out on stainless and ferritic steel flat sheets, including Ø1 × length 5 mm open rode-size defects. Good defect detection was observed for all sensors tested, confirming their compatibility with industrial ECT equipment.

## 5. Perspectives and Conclusions

Most daily used non-destructive testing methods (ultrasound, electromagnetic, etc.) give potential access to some structural health parameters and their evolution in time. Unfortunately, those features are rarely exploited. Ultrasounds can, for example, quantify fundamental mechanical characteristics (Young modulus, density, etc.) that might evolve in time as an image of an increasing degradation state. Similarly, ECT is an accurate method to detect fiber orientation, density, and delamination in carbon fiber composites. The most significant technological issue in structural health monitoring development lies in the repeatability of the measurement conditions. Portable sensors currently used in maintenance services are almost impossible to set in reproducible conditions. This issue leads to measurement variations, equally attributed to the experimental environment or the tested specimen evolutions. The solution promoted in this article involves printing the NDT sensors directly on the part to be monitored. The printed sensors provide significant information, including the evolution of fundamental parameters. The repeatability issue is solved, and measurement variations can only be attributed to changes in the targeted properties.

The ECT-printed sensors developed in this work were significantly flat (thickness < 100 μm). They can be implemented non-intrusively in structural parts, including components under high mechanical and thermal stresses. Such extreme conditions are frequently found in the military (missile, space propellant), transportation (aeronautics, rail), and energy industries. A vast experimental campaign was reported in this manuscript and concluded with the credibility of the innovative method. The excellent detection capability of the resulting sensors was noticed. Both the dispenser- and the screen-printed methods can be used equally successfully. The best results were obtained with a combination of ink supplied by the Dupont company. Please note that endurance tests in high-temperature conditions are still ongoing. Their results will be communicated later.

Multiple perspectives can be associated with this work:–Figure 1 in the Introduction depicts a mechanical component with a critical part fully paved with printed sensors. Such a mesh of sensors can be used to detect weaknesses in hazardous or inaccessible environments. Controls can be fast and automated. Still, as far as we know, it has never been tested practically and would constitute an exciting perspective.–First tests done on specimens of different compositions revealed good discrimination by the sensors. Still, these results must be confirmed in the context of aging and structural health monitoring.–In this study, the detection capability was validated by detecting a small diameter hole with a permanent printed sensor. Different defect types (nature, size, etc.) were also observed with the flexible sensors and commercialized eddy current equipment but not with the permanent sensors. They will be tested in the future.–The compatibility and adaptability of the printed sensor to industrial equipment were validated with the permanent sensors. These sensors showed good adhesion to the tested parts and electromagnetic responses comparable to portative sensors. Still, structural monitoring will require additional developments to ensure the long-term viability of these devices.

## Figures and Tables

**Figure 1 sensors-23-08345-f001:**
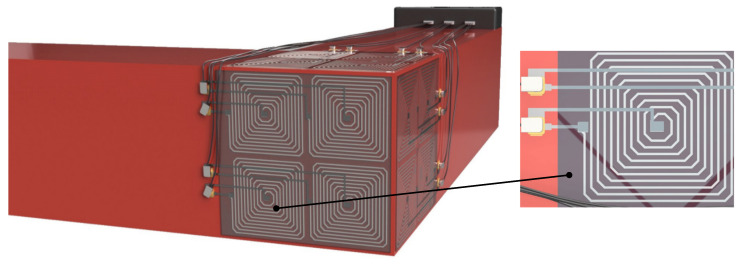
ECT planar coil sensors printed on a mechanical part.

**Figure 3 sensors-23-08345-f003:**
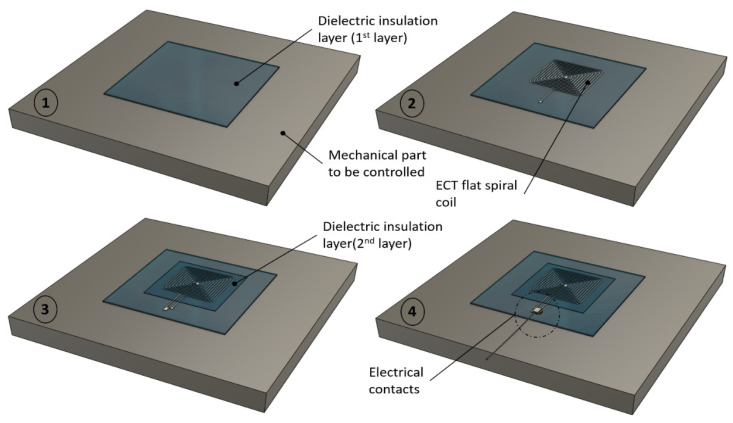
Multilayer printing process.

**Figure 4 sensors-23-08345-f004:**
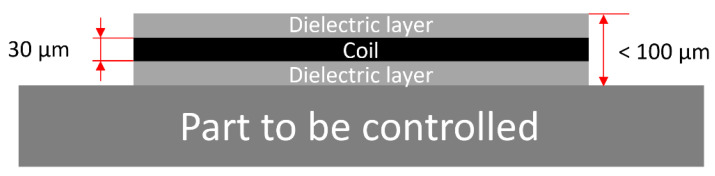
Printed eddy current sensor: typical thicknesses.

**Figure 5 sensors-23-08345-f005:**
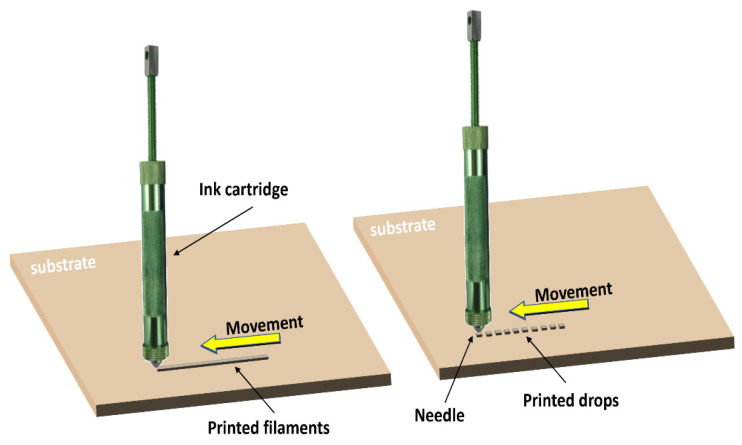
Schematic illustration of the dispenser-printing process. The ink is deposited in continuous filaments or distributed drops to form desired patterns [30].

**Figure 6 sensors-23-08345-f006:**
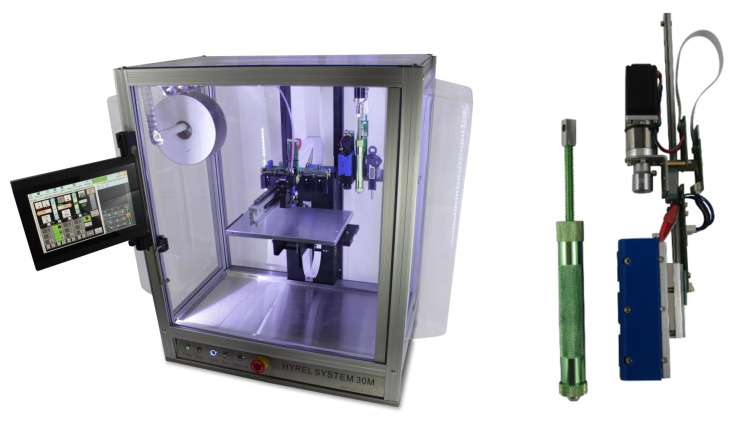
Hyrel system 30M 3D printer and VOL-25 print head.

**Figure 7 sensors-23-08345-f007:**
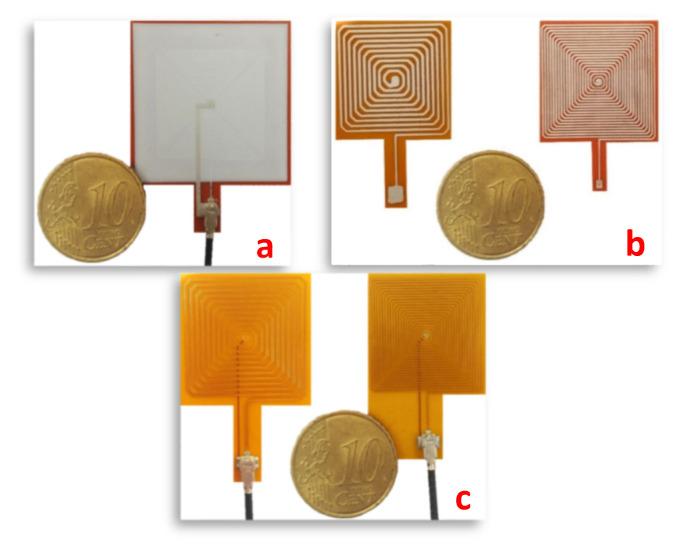
(**a**) 20-turn planar spiral coil sensor obtained with the screen-printing method. (**b**) 10- and 20-turn planar spiral coil sensors obtained with the dispenser-printing technique. (**c**) 10- and 20-turn planar spiral coil PCB-Flex sensors.

**Figure 8 sensors-23-08345-f008:**
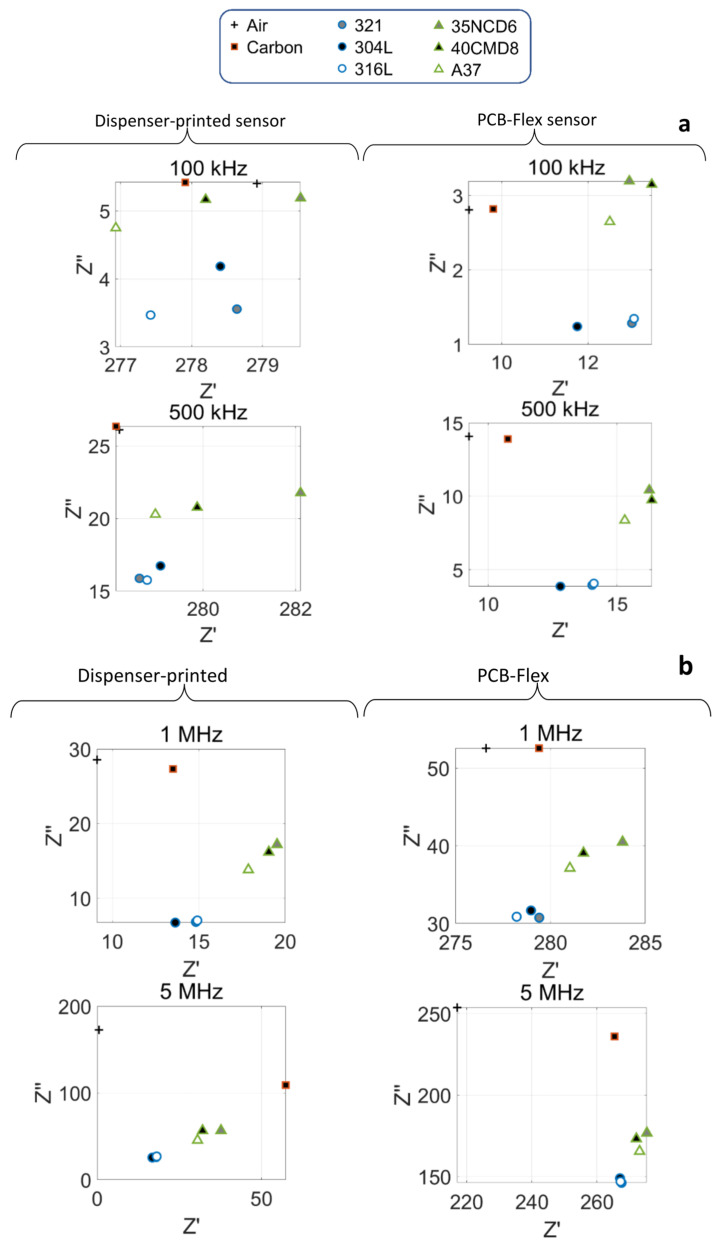
(**a**) Dispenser-printed and PCB sensors impedance characterization in the low-frequency range and for different natures of tested parts. (**b**) Same characterizations in the high-frequency range.

**Figure 9 sensors-23-08345-f009:**
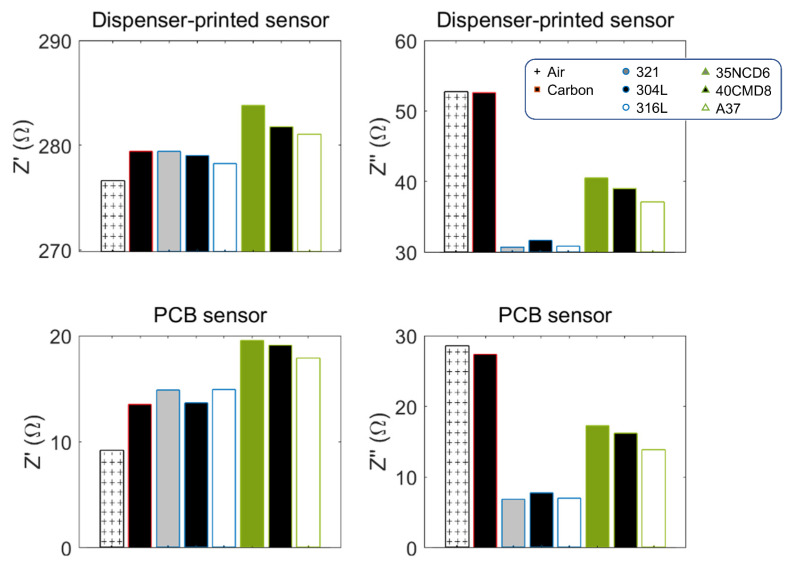
1 MHz detailed impedance comparisons.

**Figure 10 sensors-23-08345-f010:**
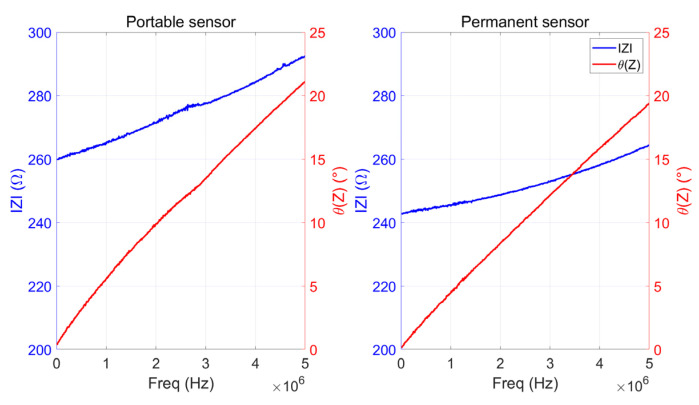
Impedance variations for a portable (**left**) and a permanent (**right**) sensor.

**Figure 11 sensors-23-08345-f011:**
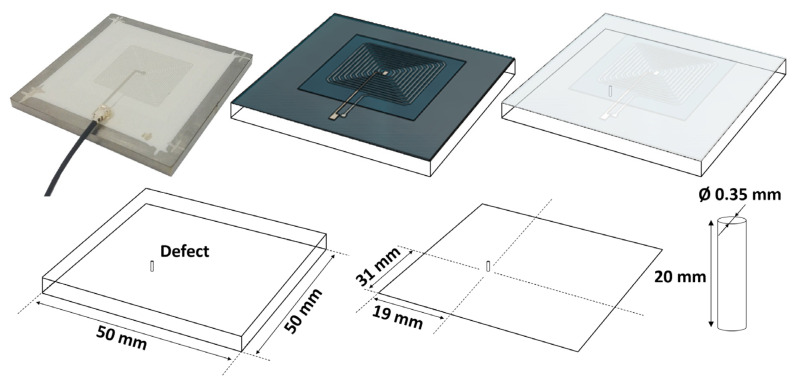
Hole defect position and geometrical information.

**Figure 12 sensors-23-08345-f012:**
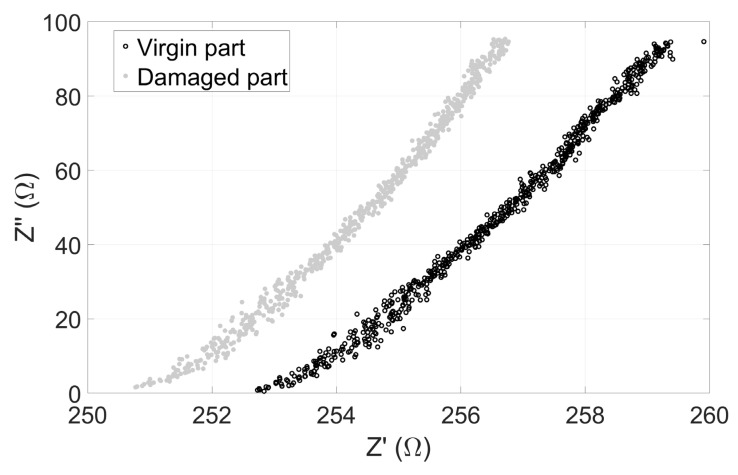
Comparison between the virgin and the damaged part impedance responses (*f* Є [100–5 × 10^6^] Hz).

**Table 1 sensors-23-08345-t001:** 10-turn spiral coils characterization results far from conductive parts.

Frequency	Dispenser Printing			Screen Printing			Flex PCB		
	R (Ω)	Lω (Ω)	Q	R (Ω)	Lω (Ω)	Q	R (Ω)	Lω (Ω)	Q
**10 kHz**	12.49	0.076	0.0061	24.99	0.087	0.0034	1.23	0.092	0.074
**100 kHz**	12.48	0.680	0.054	25.08	0.919	0.036	1.25	0.932	0.745
**500 kHz**	12.52	3.09	0.247	25.04	4.23	0.169	1.28	4.28	3.33
**2.5 MHz**	12.72	60.70	0.476	25.15	8.32	0.330	1.34	8.39	6.24
**1 MHz**	13.95	14.60	1.046	25.72	20.83	0.809	1.52	20.74	13.61
**5 MHz**	17.34	26.41	1.522	27.59	43.48	1.57	1.72	41.06	23.85

**Table 2 sensors-23-08345-t002:** 10- and 20-turn flex PCB spiral coil characterization results far from conductive parts.

Frequency	10 Turns			20 Turns		
	R (Ω)	Lω (Ω)	Q	R (Ω)	Lω (Ω)	Q
**10 kHz**	1.23	0.092	0.074	3.13	0.253	0.080
**100 kHz**	1.25	0.932	0.745	3.20	2.54	0.795
**500 kHz**	1.28	4.28	3.33	3.31	11.63	3.51
**2.5 MHz**	1.34	8.39	6.24	3.45	23.14	6.69
**1 MHz**	1.52	20.74	13.61	4.15	59.45	14.30
**5 MHz**	1.72	41.06	23.85	6.59	139.7	21.20

**Table 3 sensors-23-08345-t003:** Exhaustive list of the tested specimens.

Carbon Composite	Carbon
Stainless Steel	304L
	316L
	321
Ferritic Steel	35NCD16
	A37
	40CMD8

## Data Availability

Data are available on request due to privacy/ethical restrictions.

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
