# Peer review of "Printed Eddy Current Testing Sensors: Toward Structural Health Monitoring Applications"

_sensors, 2023, doi:10.3390/s23198345_

Round 1
Reviewer 1 Report
The manuscript is well organized and meaningful to the related filed. Based on the fluent expression and scientific significance of the manuscript, I recommend an acceptance after minor revisions.
1\Table 1 and 2 are not illustratted rightly.
2\ The abstract needs to introduce more key data of the main results.
3\Figure 1 is very abrupt, it is recommended to reduce and highlight the key parts.
Reviewer 2 Report
The Manuscript ID: sensors-2615192 ‘Printed eddy current testing sensors: Toward structural health monitoring applications’ is in fact novel attempt and will be a good addition to the field of printed sensors. The applied side is also clear. However, there are several major flaws which need to be addressed before acceptance. Very important aspects are not considered for this manuscript.
1. Introduction section must be re-written focusing the need and novelty of these printed sensors. This will enhance the presentation of this manuscript.
2. References, especially introduction section must appear from 2020-2024. The added references have least relevance with the presented points.
3. Big number of figures is actually disturbing the manuscript quality.
4. Similarly, haphazard discussion in each section must be revised carefully with clear sentences.
5. Please combine Fig. 2 & 3.
6. Please combine Fig. 8, 9, & 10 for better presentation.
7. There is no need of Fig. 13. It is already given in Fig. 15.
8. Fig. 16, must be revised with more clear presentation of data with better bullets, or present in Table form.
9. Section 5 must be corrected as ‘Perspectives and Conclusions’. This section needs to be re-written pointing out the main results of this work as well as observations. Presently, it is written so haphazardly and does not portray the worth of this important study.
10. Language and grammar need serious attention. Long sentences must be revised throughout the manuscript and several sentences are written carelessly. Can be presented in fluent language. e.g., line 169 “Each geometry shows pros and cons: circular sensors are promoted for crack detection” and so on.
Please see last point in comments.
Reviewer 3 Report
The authors described a solution that involves printing the NDT sensors directly on the part to be monitored, and the printed sensors can provide a significant amount of information. As a whole, it is well organized and can be considered after a minor revision:
(1) In the introduction section, the authors are suggested to briefly introduce the development history of this solution, and emphasize the novelty of the current work.
(2) The performance of this solution had better be compared with other NDT methods.
Reviewer 4 Report
Introduction: "However, the cost and size of these measurement methods and the need for an operator to carry out these controls hinder their practical use and overall development in the framework of material and structural health monitoring." In my opinion, this is too strictly worded. As it is also described in your Ref 4, e.g. Acoustic Emission has been used for structural monitoring since the 1970s and is state of the art for e.g. piplines and pressure valves.
Design rules: Z' and Z'' must be interchanged in equation 1.
Please add units in Tables 1 and 2.
Please add the coil manufacturing process in the heading of Table 2.
What conclusions were drawn from the results in Figures 11 and 12?
I miss a detailed discussion of the figures presented. A discussion with literature references is completely missing.
Round 2
Reviewer 2 Report
I think the manuscript is modified enough to publish now.